# Fabrication, Microstructure and Colloidal Stability of Humic Acids Loaded Fe_3_O_4_/APTES Nanosorbents for Environmental Applications

**DOI:** 10.3390/nano11061418

**Published:** 2021-05-27

**Authors:** Lyubov Bondarenko, Erzsébet Illés, Etelka Tombácz, Gulzhian Dzhardimalieva, Nina Golubeva, Olga Tushavina, Yasuhisa Adachi, Kamila Kydralieva

**Affiliations:** 1Engineering Department, Moscow Aviation Institute (National Research University), 125993 Moscow, Russia; dzhardim@icp.ac.ru (G.D.); kydralievaka@mai.ru (K.K.); 2Aerospace Department, Moscow Aviation Institute (National Research University), 125993 Moscow, Russia; solgtu@gmail.com; 3Department of Food Engineering, University of Szeged, H-6720 Szeged, Hungary; Illes.Erzsebet@chem.u-szeged.hu; 4Soós Ernő Water Technology Research and Development Center, University of Pannonia, H-8800 Nagykanizsa, Hungary; tombacz@chem.u-szeged.hu; 5Laboratory of Metal Containing Polymers, Institute of Problems of Chemical Physics, 142432 Chernogolovka, Russia; nd_golubeva@mail.ru; 6Faculty of Life and Environmental Sciences, University of Tsukuba, Ibaraki 305-8577, Japan; adachi.yasuhisa.gu@u.tsukuba.ac.jp

**Keywords:** magnetite nanoparticles, silica coating, humic acid loading, pH-dependent surface charging, colloidal stability

## Abstract

Nowadays, numerous researches are being performed to formulate nontoxic multifunctional magnetic materials possessing both high colloidal stability and magnetization, but there is a demand in the prediction of chemical and colloidal stability in water solutions. Herein, a series of silica-coated magnetite nanoparticles (MNPs) has been synthesized via the sol-gel method with and without establishing an inert atmosphere, and then it was tested in terms of humic acids (HA) loading applied as a multifunctional coating agent. The influence of ambient conditions on the microstructure, colloidal stability and HA loading of different silica-coated MNPs has been established. The XRD patterns show that the content of stoichiometric Fe_3_O_4_ decreases from 78.8% to 42.4% at inert and ambient atmosphere synthesis, respectively. The most striking observation was the shift of the MNPs isoelectric point from pH ~7 to 3, with an increasing HA reaching up to the reversal of the zeta potential sign as it was covered completely by HA molecules. The zeta potential data of MNPs can be used to predict the loading capacity for HA polyanions. The data help to understand the way for materials’ development with the complexation ability of humic acids and with the insolubility of silica gel to pave the way to develop a novel, efficient and magnetically separable adsorbent for contaminant removal.

## 1. Introduction

The magnetite nanoparticles (MNPs) have a wide range of applications, from the magnetic separation of technical media [1,2,3] to the preparation of polymer suspensions for biomedicine [4,5,6,7,8] and the life sciences [9]. The modification of MNPs with different species of stabilizing agents leads to producing monodispersed nanopowders and to supplying specific interactions with the molecules due to appropriate functional groups needed in various applications. Humic substances (HS), natural high-molecular-weight organic compounds [9], seem rather prospective to coat the nanoparticles, thereby preventing aggregation through either electrostatic or steric stabilization [10], due to an abundance of carboxylic (−COOH, −COO−) and phenolic (−OH) functional groups, HS use is not only limited to preventing aggregation of nanoparticles [10,11,12], but it suggests that derivatives of humic acids could play a role in processes of water cleaning from toxic metals [13,14,15] and organic contaminants [16,17].

Although HAs have a multifunctional character and thus constitute a suitable environment for various species [18], and despite the apparent simplicity of magnetite NPs synthesis in humic acids matrix [10,12], some of its drawbacks, such as its high solubility, restrict the utilization of HA as a substrate [19]. The resulting coulombic, hydrogen and other bonds between Fe_3_O_4_ and HA seemed to provide low protection to prevent oxidation of NPs, especially in ambient conditions (UV, biota or groundwater chemistry, i.e., pH, redox potential, dissolved oxygen, alkalinity). In particular, after their release into the environment, magnetite-humic NPs undergo various transformations via interactions with different geochemical and biological components, which ultimately influence their behavior and potential toxicity.

For this reason, immobilization of HA to a suitable, solid support with good resilience is deemed important in order to take the benefit of its multifunctional character [20]. We believe it is a good tentative idea to combine the multifunctional character of HA and the magnetic properties of iron oxide particles through the covalent immobilization of HA [21].

The use of 3-aminopropyl-triethoxysilane (APTES) as a derivative of silica to form strong covalent bonds with the Fe_3_O_4_ MNPs and functionalize NPs surfaces seems effective due to its terminal amino groups, stability under acidic conditions and inertness to redox reactions [22]. In a number of works, successful functionalization and wide applications of Fe_3_O_4_ nanoparticles with APTES were reported [23,24,25].

It is well known that the variation of the synthesis parameters (atmosphere, time, temperature, pH, the concentration of components, etc.) influences the final nanoparticles and their physical properties [26]. However, the most important parameter in the preparation process is the atmosphere, especially iron oxide nanoparticles, which are known to be oxygen-sensitive. However, limited works are aimed at studying the influence of oxygen on magnetite nanoparticles [26].

In view of the above-mentioned aspects, a series of sol-gel syntheses of the magnetite nanoparticles coated by APTES in different atmospheres (argon and ambient) was accomplished and tested in terms of HA loading in this work. Unlike the work [23], our APTES-coated particles were synthesized to compare without establishing an inert atmosphere during the synthesis in order to (i) simulate ambient conditions in terms of the effect of dissolved oxygen on the microstructure of MNPs, and (ii) to consider a low-cost NH_2_-silica-functionalization route. Due to the positive charge of NH_2_-groups of APTES, the electrophoretic light scattering (ELS) and the value of the zeta potential of NPs seem to be informative enough to determine the loading potential for HA polyanions. Thus, the formulation of the above multilayered, hybrid materials may lead to new materials with the complexation ability of humic acids, the magnetism of MNPs and with the insolubility, thermal and chemical stability characteristics of silica gel. This type of material utilizes immobilized humic acids for any application, including removing contaminants from water and/or other solvents and medical uses, while avoiding humic acid loss by solubilization. In order to be able to tune the colloidal stability of the magnetically separable multifunctional adsorbent to be used in water purification in the future, the pH-dependent charge state and colloidal stability of different products have to be characterized.

## 2. Materials and Methods

### 2.1. Synthesis of Fe_3_O_4_ MNPs

The bare Fe_3_O_4_ MNPs were prepared by the coprecipitation method described in [12]. Briefly, 7.56 g of FeCl_3_·6H_2_O and 2.78 g of FeCl_2_·4H_2_O were dissolved in 70 mL of H_2_O, which was then added to 40 mL of 25% solution of ammonium hydroxide at 50 °C under argon flow with vigorous stirring. The obtained Fe_3_O_4_ nanoparticles were washed with ultrapure water five times to remove the synthesis residues, then dried at 70 °C under vacuum.

### 2.2. Synthesis of Fe_3_O_4_/APTES MNPs

The synthesis of the two composites of silica-coated magnetite nanoparticles was carried out by the well-known Stöber method [27]. The previously synthesized magnetite nanoparticles were used as cores to be coated with SiO_2_. We used 3-aminopropyltriethoxysilane (APTES, 98% purity, Sigma-Aldrich Chemie GmbH, Steinheim, Germany) as the NH_2_-silica precursor. The two composites of NH_2_-silica functionalized silica magnetite nanoparticles are identified as Fe_3_O_4_/APTES(Ar) and Fe_3_O_4_/APTES (air). In the synthesis of the Fe_3_O_4_/APTES(Ar) and Fe_3_O_4_/APTES (air) nanocomposites, argon and ambient (air) atmosphere were used, respectively. According to [28], 3.21 g of Fe_3_O_4_ MNPs was dispersed in 150 mL of ethanol/water (volume ratio, 1:1) solution. Then 13.6 g of APTES was added to the solution under argon or air atmosphere at 40 °C for 2 h. The molar ratio of APTES to Fe_3_O_4_ was used as 4:1. The room temperature, cooled solution of the formulated Fe_3_O_4_/APTES(Ar) and Fe_3_O_4_/APTES (air) MNPs was separated with a magnet (Nd, 0.3 Ts). All subsequent silanization reactions were performed in 1/1 mixtures of ethanol/water and extensively washed with water before analysis or further use. This was done to avoid the occurrence of nonspecifically bound silane to the particle surfaces. Finally, the Fe_3_O_4_/APTES(Ar) and Fe_3_O_4_/APTES (air) samples were vacuum-dried at 70 °C for 2 h.

### 2.3. Humic Acids Characterization

The sample of HA was prepared from sodium humate (Powhumus, Humintech, Grevenbroich, Germany). The low H/C atomic ratio (0.85) suggests a high content of aromatic structures in the HA preparation. The total acidity of the sample was 5.3 mmol/g of acidic COOH and OH-groups, the weight-average molecular weight Mw was 9.9 kD.

In the total, three types of samples were formulated: Fe_3_O_4_, Fe_3_O_4_/APTES(Ar) and Fe_3_O_4_/APTES (air) to analyze microstructure, surface parameters and estimate loading capacity towards HA.

### 2.4. Characterization of Samples

The phase composition and particle size of the MPNs samples were determined by X-ray diffraction analysis (XRD) in Bragg–Brentano geometry using a Philips X-pert diffractometer (Cr-Kα radiation, λ = 2.29106 Å, Philips Analytical, Eindhoven, The Netherlands). The full width at a half maximum (FWHM) of all the reflections was used for the particle size determination with the Scherrer equation. In order to quantify the oxidation progress, the (440) reflection was fitted with five different functions in Origin 2019 Pro (OriginLab Corporation, Northampton, UK). The morphology of the nanomaterials was investigated by scanning electron microscopy (SEM) using a SUPRA 55VP-32-49 microscope (Coventry, West Midlands, UK). The average particle diameter of the synthesized MNPs was determined by measuring the diameters of more than 100 particles using ImageJ software ± Standard deviation. The data were fitted with Lognormal functions (95% confidence interval) using Origin 2019 Pro (OriginLab Corporation, Northampton, UK). Infrared spectra were recorded in the 4000–400 cm^−1^ range on a Fourier transform infrared spectrometer (FTIR IR-200, ThermoNicolet, Waltham, MA, USA). The samples were pressed into tablets with *Ø* = 13 mm with spectrally pure KBr (pressing force of 6 tons) at the rate of 1 mg of the powder of the analyte per 150 mg KBr.

Dynamic light scattering (DLS) measurements were conducted with a NanoZS apparatus (Malvern Panalytical Ltd., Malvern, UK) at a wavelength of 633 nm with a solid-state He–Ne laser at a scattering angle of 173° at 25 °C. For the DLS analysis, each sample was diluted to approximately 0.1 g/L. Since the colloidal stability of the samples varies with the solution conditions, the particles may aggregate, all of which were measured at a given kinetic state achieved by 10 s of ultrasonication followed by 100 s of standstill. The experiments were carried out in a disposable zeta cell (DTS 1070). The range of pH was between ~3 and ~10. The values of pH before and after the study on the device were measured. Subsequently, the value of pH after measurement by the method of dynamic light scattering was used. The experiments were performed at constant ionic strengths 0.01 M set by KCl. The effect of HA loading on the sorption capacity of MNPs was tested.

ANOVA was used for the analysis of statistically significant variances within and between the zeta potentials and hydrodynamic sizes. The degree of statistical significance of the results was calculated in the R-studio application. The programs were created in the programming language R (inter-group statistical significance was fixed at *p* < 0.05). All data were fitted with different functions using Origin 2019 Pro with R^2^ ≥ 0.99.

## 3. Results

### 3.1. Microstructure of Fe_3_O_4_/APTES MNPs

XRD analysis and SEM. The crystalline structures of the nanoparticles were identified with XRD analysis (Figure 1). Powder XRD yields the crystal structure, and with analysis of crystal defects, the diameter of nanoparticles can be estimated. Furthermore, it is possible to differentiate between the phases magnetite and maghemite even though they have a very similar spinel and distorted spinal structure, respectively [26]. The reflections were fitted with the five most popular models: Gauss function, Lorenz function, Voigt function, Pseudo-Voigt function and PearsonVII function in OriginPro 9.1. According to R^2^ and χ^2^, data for Fe_3_O_4_ and Fe_3_O_4_/APTES (air) samples were most accurately fitted by the Pseudo-Voigt function (R^2^ = 0.99 and χ^2^ = 55.4), while data for Fe_3_O_4_/APTES (Ar) were by the PearsonVII function (R^2^ = 0.99 and χ^2^ = 49.9). The XRD patterns are similar for all samples. They can be interpreted as a face-centered cubic (fcc) lattice with the parameters of 8.381(3), 8.378(9) and 8.360(3) Å for the Fe_3_O_4,_ Fe_3_O_4_/APTES (Ar) and Fe_3_O_4_/APTES (air) samples, respectively (Table 1). Alkoxysilanes can give a wide peak in the diffraction patterns; in this case, a wide peak can be attributed to the sample preparation effect.

The lattice parameters determined for all samples formulated in this study are smaller than those reported for magnetite 8.3873 Å (COD#9009769) (8.396–8.400 Å (ICDD–PDF 19–629)) but larger than those for maghemite 8.3364 Å (COD#2300617) (8.33–8.34 Å (ICDD–PDF 39–1346)). A plausible explanation of this phenomenon could be the process of partial oxidation of Fe^2+^ during drying and storage, resulting in the non-stoichiometric Fe_3-δ_O_4_ formation where *δ* can range from zero (stoichiometric magnetite) to 1/3 (completely oxidized) [29]. For magnetite with an ideal Fe^2+^ content (assuming the Fe_3_O_4_ formula), the mineral phase is known as stoichiometric magnetite (*x* = 0.50). As magnetite becomes oxidized, the Fe^2+^/Fe^3+^ ratio (Formula 1) decreases (*x* < 0.50), with this form denoted as nonstoichiometric or partially oxidized magnetite [29]. The stoichiometry can easily be converted to the following relationship:(1)$$x=\frac{{\mathrm{Fe}}^{2+}}{{\mathrm{Fe}}^{3+}}=\frac{1-3\delta }{2+2\delta }$$

Finally, the composition of the crystalline component of the samples can be assigned as follows: Fe_2.93_O_4_, Fe_2.92_O_4_ and Fe_2.81_O_4_ for the Fe_3_O_4,_ Fe_3_O_4_/APTES (Ar) and *Fe*_3_O_4_/APTES (air) samples, respectively (Table 1). A decrease in the magnetite content in the sample with a silica shell can be associated with the oxidation of magnetite in the synthesis process. He and Traina [30] showed that magnetite particles treated with dilute NaOH solution to some extent were transformed to maghemite.

The coherent-scattering region size was derived from powder XRD data by Scherrer’s method. Changes in the size of nanoparticles upon coating studied by SEM and XRD correlated. While the spherical particle shape remained constant for all modification routes, a slight particle growth can be observed (Figure 2). The bare magnetite particles exhibit a median particle diameter of 17.1 and 32.1 nm for XRD and SEM analysis, respectively. Such magnitudes for the magnetic iron oxide nanoparticles are known to be for superparamagnetic MNP_S_ with a high saturation magnetization and a high specific surface area [31,32]. The diameter of the Fe_3_O_4_/APTES (Ar) particles changed to 20.5 and 24.2 nm compared to bare MNPs, while the Fe_3_O_4_/APTES (air) particles demonstrated a diameter gain to 18.5 and 23.3 nm for XRD and SEM analysis, respectively. The crystallite size increase of APTES-modified NPs, in comparison with bare NPs, happened when the particles were submitted to the thermal sol-gel treatment, which resulted in the modification of their physical properties [24]. This is in agreement with others papers, which showed an increase in crystallite size from 10 to 13 nm [33] and 13.4 to 18.4 nm [23] after amino-silica functionalization. According to the coefficient of variation (CV) and standard deviation value (σ) of samples for SEM, Fe_3_O_4_/APTES (air) has a smaller size distribution (10.5%, 3.1) than Fe_3_O_4_ (13.5%, 4.3) and Fe_3_O_4_/APTES (Ar) (11.6%, 2.8). Despite that, all samples are considered as polydisperse according to [34]. A CV of 10.5% and an SD of 3.1 indicate the homogeneity of the Fe_3_O_4_/APTES (air) NPs crucial for the final performance of the surface-activated material. Thus, the synthesis atmosphere (argon or air) only moderately affects the particle size, but we observed an influence on the lattice parameters of magnetite leading to its deeper crystal defects. The content of stoichiometric magnetite in the air-synthesized route sample decreased from 75.8 to 42.2%.

FTIR spectra. The FTIR spectra of Fe_3_O_4,_ Fe_3_O_4_/APTES(Ar) and Fe_3_O_4_/APTES (air) nanoparticles are compared in Figure 3. The FTIR spectrum of Fe_3_O_4_ exhibits infrared absorption bands at 424, 592 and 628 cm^−1^, that can be assigned to the Fe-O stretching mode of the tetrahedral and octahedral sites for the band at 592 cm^−1^ and the Fe-O stretching mode of the octahedral sites for the band at 424 cm^−1^ [35]. For the Fe_3_O_4_/APTES(Ar), the bands 396 and 592 are ascribed to the Fe-O bond of Fe_3_O_4_ [35], while 620 cm^−1^ can be attributed to the Fe_2_O_3_ or Fe-O-Si bonds as well. The bond 464 cm^−1^ observed on Fe_3_O_4_/APTES(Ar) can be ascribed to the stretching and deformation vibrations of SiO_2_, reflecting the coating of silica on the magnetite surface. For the Fe_3_O_4_/APTES (air), the characteristic absorption bands of the Fe–O bond shifted to a higher wavenumber (636 cm^−1^) compared to the uncoated MNPs (at 628 cm^−1^) and can be attributed to Fe–O–Si bonds according to the results of Bini et al. [24]. On the other hand, a band at 636 cm^−1^ of Fe_3_O_4_/APTES (air) can be assigned to Fe-O of maghemite as indicated in [35]. The band at 416 and 444 cm^−1^ of Fe_3_O_4_/APTES (air) and 592 cm^−1^ of Fe_3_O_4_ and Fe_3_O_4_/APTES (air) can be associated with Fe-O bonds. The presence of Fe-O-Si bonds for the Fe_3_O_4_/APTES (air) cannot be seen in the FTIR spectrum because it likely appears at around 584 cm^−1^ and therefore overlaps with the Fe–O vibration of magnetite NPs [36].

Surface modification by APTES is a complex process since it does not include a single mechanism and many different intermediates are possible [37]. The FTIR spectra demonstrate the absorption bands for the silanol (Si–O–H) and siloxane (Si–O–Si) groups. The band 1008 cm^−1^ of Fe_3_O_4_/APTES (air) is attributed to the Si-O-Si groups, respectively [36,38], confirming the adsorption of APTES on the MNP surface. The band around 992 cm^−1^ of Fe_3_O_4_-APTES(Ar) is assigned to the Si-O [39] or Si-OH [38] groups. At the same time, White [40] reported that bands 1008 or 992 cm^−1^ could be attributed to the SiO bond of the SiOH-group, which H-bonded to NH_2_-group (Figure 3, case 4). White also reported on the formation of so-called “arched” structures, in which the amino group interacts with the OH-group Si-OH or Fe-OH. According to [41], band 1392 cm^−1^ of the Fe_3_O_4_ spectra can be attributed to the OH-groups. The absence of bands around 2979 cm^−1^ for Fe_3_O_4_/APTES (air) and Fe_3_O_4_/APTES (Ar) indicates the absence of OCH_3_-groups [41].

There are three different explanations of ~1600 and ~3400 cm^−1^ bands: free NH_2_, bonding of N-H or OH-groups. Broad bands at 1628 and 3416 cm^−1^ for Fe_3_O_4_ and 3416 cm^−1^ for Fe_3_O_4_/APTES (air) could be identified as OH-stretching vibration. At the same time, 1624 and 1616 cm^−1^ for Fe_3_O_4_-APTES(Ar) and Fe_3_O_4_/APTES (air), respectively, can be identified as the N–H stretching vibration, which indicates as the free amino group [42]. A small band at 3448 cm^−1^ for Fe_3_O_4_/APTES(Ar) can be ascribed OH-stretching vibration of the OH-group or amino-group, in accordance with the small fraction of the alkoxysilane on the surface layer [42].

The intensity of the APTES bands for the MNPs modified in Ar and air was relatively weak. The relative insensitivity of IR analysis with respect to surface modification of the nanoparticles with APTES was likely to render it incapable of detecting what could have been subtle changes with respect to surface composition resulting from these experiments [43]. These observations line up with the fact that the surface silane layer represents a small amount of the material’s mass in comparison with its bulk of magnetite. Surprisingly, the more intense bands were obtained when the reaction was performed in air, suggesting that much more APTES had been deposited to the nanoparticles’ surface. This could be due to the higher density of surface OH-groups nm^−2^ of the maghemite, of 9.8, in comparison to those of the magnetite, of 5.2 [44], leading to a more effective silanization due to the interchanging of hydroxyl groups of nanoparticles with APTES moieties. However, there are other data from Jolstere et al. [45] suggesting that the number of proton-active surface sites is 1.5 ± 0.8 sites nm^−2^, which is 1.5 times larger than the maghemite (0.99 ± 0.05) [45]. This rather high value of active surface sites of magnetite has been questioned by Tombacz et al. [46].

On the other hand, increased intensity for the OH-groups region can be a result of the uncompleted polymerization of APTES. However, it should also be stated that in the case of successful polymerization, there will be OH-groups on the surface; therefore, an increase in intensity cannot directly be evidence of incomplete polymerization.

Thus, the oxidative environment does not influence the silanization in the same way that the phase iron oxide nanoparticles do. This circumstance allows for modifying APTES without the establishment of an inert atmosphere, i.e., in low-cost conditions. In the spectra for both samples of Fe_3_O_4_/APTES, the peaks for the functional groups and silane binding assigned to the Si-O groups confirm the introduction of APTES to the surface of magnetite NPs and NH_2_ groups on the surface of APTES.

### 3.2. Characteristics of Surface Charging and Hydrodynamic Size of MNPs

In the case of all the samples obtained, the core is magnetite in a maghemite shell, which means that only Fe^3+^ and OH are presented on the NPs surface (Figure 4).

With the prevalence of hydroxyl groups on the surface (from Fe-OH after unsuccessful functionalization or -C_2_H_5_OH in case of incomplete polymerization and ≡Fe-O-… + H_3_N-R bonds), the particle will have a negative charge (right part of the figure), with the prevalence of amino groups—a positive charge (left part of the figure). In accordance with the above surface scenarios, the following reactions can occur on the surface during protonation and deprotonation (Equations (2)–(7)) [42]:(2)$$\equiv \mathrm{Fe}-\mathrm{OH}+H^{+}\;\Leftrightarrow \;\equiv \mathrm{Fe}-{\mathrm{OH}}_{2}^{+}$$
≡Fe-OH + OH^−^ ⇔ ≡Fe-O^−^ + H_2_O(3)
(4)$$R-\mathrm{OH}+H^{+}\;\Leftrightarrow \;R-{\mathrm{OH}}_{2}^{+}$$
R-OH + OH^−^ ⇔ -RO^−^ + H_2_O(5)
(6)$$R-{\mathrm{NH}}_{2}+H^{+}\;\Leftrightarrow \;R-{\mathrm{NH}}_{3}^{+}$$
R-NH_3_^+^ + OH^−^ = R-NH_2_ + H_2_O(7)

The presence of the amine function should shift the IEP towards higher pH values, as the intrinsic pKa of the aminopropyl is 9.8, and the amino groups should be fully protonated at pHs less than about 9 [47]. Depending on the effective surface concentration of the amine groups and the surface concentration of the free silanol groups, the electrokinetic surface potential could vary from slightly positive to strongly positive due to the charge balancing effect of the deprotonating surface silanols [47].

#### 3.2.1. Effects of APTES and Oxidation

Due to the positive charge of NH_2_-groups of APTES, the method of electrophoretic light scattering (ELS) and the value of the zeta potential of NPs seem to be informative to determine the loading potential for the negative charged HA. The adsorption of humic acids on magnetite nanoparticles focusing on surface charges of magnetite was first reported by Illes and Tombacz [48]. The adsorption studies through surface charges study revealed that the amount of loaded HA is influenced by the pH. This study is of interest for improving the humic preparations to be available to immobilize on the mineral surfaces and therefore prolonged migration and stability.

The pH-dependent charge state of particles measured as zeta potential is presented in Figure 5 for the Fe_3_O_4_, Fe_3_O_4_/APTES(Ar) and Fe_3_O_4_/APTES (air) nanoparticles. It is evident from Figure 5 that the profiles of zeta potential curves for APTES-based NPs were dependent on the synthesis way (Table 1). Different synthesis conditions lead, therefore, to different surface charges. When examining the profiles of zeta potential between Fe_3_O_4_/APTES NPs samples synthesized in argon and air atmosphere in Figure 5, it is evident that a significant difference is observed practically in the entire pH range.

The isoelectric point (IEP) of Fe_3_O_4_ NPs (Figure 5) was found to be 6.3, which is in accordance with values in the literature [49]. The IEP corresponds to the pH regime where the net charge on the NPs reaches zero [50]. The reactions of surface ≡Fe–OH sites of magnetite, which can lead to the formation of positive (≡Fe–${\mathrm{OH}}_{2}^{+}$) and negative (≡Fe–O^−^) surface charges, were observed [48]. In comparison, with Fe_3_O_4_, the presence of the amine groups should shift the IEP towards higher pH values [23]. The IEP values for Fe_3_O_4_/APTES(Ar) and Fe_3_O_4_/APTES (air) were established at pH 7.1 and 6.6, respectively. The IEP value of Fe_3_O_4_/APTES(Ar) being around pH = 7 is attributed to the presence of protonated amino groups on the SiO_2_ surface [51,52]. This is combined with the position of the zeta potential curve, whereby there are fewer negative charges in the alkaline region on the surface of Fe_3_O_4_/APTES(Ar) than on the surface of Fe_3_O_4_ and Fe_3_O_4_/APTES (air). The reactions of surface Fe_3_O_4_-Si–NH_2_ sites with H+ and OH^−^-ions lead to the formation of positive (≡Fe-O-Si-${\mathrm{NH}}_{3}^{+}$) surface charges and uncharged NH_2_-group, correspondingly. Bini [24] also reported on the presence of amino groups on the surface of the functionalized sample, which is confirmed by the shift of the isoelectric point to the alkaline region (Table 2). Various IEP values for Fe_3_O_4_/APTES (Ar) and other samples from [23,24,33] can be explained by different concentrations of the surface amino groups [53].

So, it can be assumed that the moderate shift of IEP for Fe_3_O_4_/APTES (air) (6.3 vs. 6.6) indicated the slight polycondensation extent, as shown by IR studies. However, it cannot be ignored that the bare magnetite was transformed to maghemite after functionalization in the air atmosphere. Rodriquez et al. [50] have reported that the IEP for γ-Fe_2_O_3_ NPs is located at pH = 3.22. This IEP is very low, and we suspect the lowering effect of an unidentified trace contaminant like humate, phosphate. The polycondensation with APTES shifted the IEP to 6.41, which correspond to the APTES pKa value. The γ-Fe_2_O_3_/APTES-beads are positively charged at low pH (<6.41), while negatively charged at high pH [51]. Therefore, this fact can confirm the γ-Fe_2_O_3_ functionalization with APTES in air conditions.

In the acidic region, there are more positive charges on the surface of Fe_3_O_4_/APTES (air) than those of Fe_3_O_4_ and Fe_3_O_4_/APTES(Ar). This could be indicated by the greater amount of alcohol OH groups on the surface due to incomplete polymerization and/or adsorbed amino groups being H-bonded or protonated [40]. According to [43], hydrophilic and protic solvents (like alcohols) usually accelerate hydrolysis and condensation kinetics, thus promoting the surface modification process. However, they can also compete with the silane for surface silanol groups by H-bonding. Solvent molecules can also invert the silane hydrolysis reaction or even form stable complexes with the hydrolyzed species that can lower their reactivity [43]. Organosilane (3-aminopropyl)triethoxysilane can bind to a metal oxide by adsorption or covalent bonding and through the active amino group in its structure [24].

The Fe_3_O_4_ and Fe_3_O_4_/APTES (air) samples are unstable in a narrow range of pH 5–7, i.e., near the isoelectric point, the zeta potential values in the range of other pH values are approximately +30 mV (Figure 5). This value proves the system’s stability. At the same time, the Fe_3_O_4_/APTES(Ar) sample is only stable in the pH range 3–5, and the zeta potential starts to decrease with the system becoming less and less stable. As a result, in the alkaline region with pH > 7, the sample becomes unstable, which agrees with the value of the hydrodynamic diameter in this range (Figure 5, right side).

The situation for hydrodynamic diameter parameter reflected a more complicating process in the case of surface modification of nanoparticles in the air atmosphere (in the presence of dissolved oxygen). Probably, particle interparticle cross-linking through siloxane bridges may appear in air conditions as a consequence of uncontrolled reaction conditions. Figure 5 shows the DLS measurement of Fe_3_O_4,_ Fe_3_O_4_/APTES (air) and Fe_3_O_4_/APTES (Ar) particles in water. The Fe_3_O_4_ and Fe_3_O_4_/APTES (air) sample show a significant increase in average particle size over the range of pH~6–8 (490 and 650 nm, respectively), proving a pronounced aggregation near the pH of IEP, where the electrostatic repulsion between particles is negligible, and particles generally undergo fast coagulation, in accordance with [46]. The higher OH-group content was a reason for a higher hydrodynamic diameter of Fe_3_O_4_/APTES (air) compared to bare Fe_3_O_4_. In contrast with Fe_3_O_4_ and Fe_3_O_4_/APTES (air), Fe_3_O_4_/APTES (Ar) has the lesser average size in the whole range of pH, wherein we can see two plateau ranges: at 3 ≤ pH ≤ 5 with ~140 nm and 6 ≤ pH ≤ 10 with ~270 nm. The second plateau is apparently consistent with a low zeta potential (around −10 mV) at the point of IEP, and a slight increase in the hydrodynamic diameter is observed. The low zeta potential at the range 7 ≤ pH ≤ 10 and the lower hydrodynamic diameter is probably due to the presence of hydrophobic uncharged NH_2_-group on the surface of Fe_3_O_4_/APTES (Ar), which is in agreement with the data of the IR-spectroscopy.

To further evaluate the zeta potential and the size data, which are crucial for the capacitance of NP samples that are limited by the number (density) and distribution of surface groups, the NP’s ability to interact with the complementary/target molecular species—humic acids—was studied.

#### 3.2.2. Effects of HA Adsorption on MNPs Charge

The content of HA was changed to reach completely covered nanocomposites. The change in the particle charge state due to adsorption of HA on magnetite and silica-coated NPs is represented by changes in the electrokinetic potential (Figure 6) as a function of the added amount of HA at the conditions of the adsorption isotherm (pH∼5 and KCl = 0.01 M).

During the adsorption, the HA takes negative charges to the surface in excess of that necessary to neutralize the original positive charges of the magnetite and amino-silica-coated MNPs at the given pH and ionic strength. As seen in Figure 6, the electrokinetic potential decreases to zero (i.e., the surface charge of NPs is fully neutralized) when the added amount of HA reaches specific amounts. The amount of carboxylic acids at the point of charge neutralization (the zero value of the electrokinetic potential, a kind of isoelectric point—IEP) is nearly the same for Fe_3_O_4_ and Fe_3_O_4_/APTES (air), probably, due to the similarity of the surface. However, the significant difference in the sorption capacity in relation to humic acids (Figure 6) indicates the participation not only of electrostatic forces in binding. In this case, the product Fe_3_O_4_/APTES (Ar) only needs 0.005 g of humic acids to completely neutralize the charge. According to [54], these amounts are the actual moles of COO− groups linking the HA to the surface sites. Further adsorption of the HA in excess of surface charge neutralization causes charge reversal of particles.

Adsorption of HA caused zeta potential reversal and increased the absolute value of the zeta potential to −20 mV at different amounts of adsorbed HA (Table 2). While HA adsorption can increase up to some specific value (different for NPs) (g/g for plateau value in Figure 6), the absolute value of the electrokinetic potential of the particles does not increase further with an increasing amount of added HA (Table 2), probably due to adsorption saturation. The saturation with humic acids above the charge neutralization point probably indicates the participation of other types of bonds (covalent, donor-acceptor). In addition, this is also expected because of the limited increase in the values of electrokinetic mobility and due to counter ion condensation at high potentials and low electrolyte concentrations [51].

On Figure 7 the influence of different HA content is presented. Ratios as 0.0077, 0.0129, 0.0258, 0.0387, 0.0516, 0.0645, 0.0903 and 0.1161 g of HA per 1 g of Fe_3_O_4_ were used.

At certain g/g concentrations of humic acids, it is possible to achieve complete coverage of nanoparticles (Figure 8). According to [48], HA has a negative charge throughout the pH range studied here. Adsorption of polyanionic HA on iron oxide nanoparticles and amino-silica-coated induces a zeta potential reversal in the acidic region; the negative charges on nanoparticles become gradually dominant. The reach of a negative surface over the entire pH (3–10) interval indicates a complete coverage of the surface with humic acids. The maximum value of the negative zeta potential that can be achieved with an increase in the concentration of HA is 40 mV. Interestingly to note that a small amount of humic acids (around 0.007 g) is required to achieve a negative charge of −40 mV in the case of a sample Fe_3_O_4_/APTES (Ar) at pH ≥ 7. This is probably due to the conformation of humic acids, which have an expanded structure in the alkaline region, and the formation of multisite binding of HA with the Fe_3_O_4_/APTES (Ar) surface.

In the case of incomplete coating with humic acids, the zeta potential of the samples is in the range ±20 mV< zeta potential < −20 mV at pH < 7–8, which indicates the instability of the nanoparticles. Upon reaching full-coverage of HA, the zeta potential value becomes zeta potential > −20 mV in the entire pH range.

The isoelectric point offset can be used to analyze the degree of coverage of the sample with humic acids. The isoelectric point of the nanocomposites shifts towards a decrease in pH with increasing HA concentration (from ~6.2 at 0 g/g HA to ~3 at max concentration HA for Fe_3_O_4_). As seen in Figure 7, the IEP of MNPs shifts from pH 6.4 (for Fe_3_O_4_/APTES (Ar) to 7.1 and for Fe_3_O_4_/APTES (air) from 6.6) to lower pH values upon the addition of 0.007 g HA. At their higher added amounts (0.039, 0.026 and 0.065 g HA to g of Fe_3_O_4_, Fe_3_O_4_/APTES (Ar) and Fe_3_O_4_/APTES (air), Figure 7), the electrokinetic potential of the coated particles was negative in the entire range of pH studied here. The results showed that the pH-dependent stability shifts in parallel with IEP in each case; the differences are only in the amounts of HA that can completely mask the original amphoteric feature of magnetite. When the position of the isoelectric point is of pH ~3 or less, the surface is completely covered with humic acids in the case of all samples.

Covering the magnetite particles with a silica shell leads to an increase by almost one and a half times the *S_BET_* value: *S_BET_* of Fe_3_O_4_ = 100 m^2^·g^−1^, *S_BET_* of Fe_3_O_4_-silica-NH_2_ = 144 m^2^·g^−1^ [48] or from 114 m^2^·g^−1^ for Fe_3_O_4_ to 216 m^2^·g^−1^ for Fe_3_O_4_/APTES (air) [37]. This is combined well with an increase in the sorption capacity of the Fe_3_O_4_/APTES (air) sample at the entire pH range see Figure 8, pH 3 and pH 9) in comparison with bare magnetite, as was shown in this paper. At the same time, the increase in the sorption capacity with respect to humic acids for the sample subjected to oxidation may be associated with an increase in the number of OH-groups on the surface, which is in agreement with the IR spectroscopy data. The lower capacity of Fe_3_O_4_/APTES (Ar) in respect to HA is likely due to the more hydrophobic nature of the surface as a result of more effective polymerization in the controlled conditions [55].

According to the magnitude of the shift of the pH-dependent zeta potential functions, due to the sorption of an increasing amount of humic acids for all samples, we can propose an order for the decreasing sorption capacity:Fe_3_O_4_/APTES (air) ≫ Fe_3_O_4_ ≈ Fe_3_O_4_/APTES (Ar)

Figure 9 demonstrates the change in colloidal stability at pH∼5 and KCl= 0.01 M. For the polyelectrolyte-free system, the particles of Fe_3_O_4_, Fe_3_O_4_/APTES(Ar) aggregated and settled due to the low charge density and the low electrokinetic potential. The same kind of aggregation and sedimentation were observed until a certain amount of HA was added (Figure 7). At that point, the electrokinetic potential attained a value of approximately −20 mV, sufficiently high negative to keep the nanoparticles dispersed in the form of a stable, transparent colloidal sol. Samples of Fe_3_O_4_, Fe_3_O_4_/APTES (Ar) in the presence of humic acids at a concentration of 0–0.012 g become unstable, aggregated and sedimented due to the lower surface charge. The sample Fe_3_O_4_/APTES (air) remains stable even at a low concentration of humic acids (0–0.025 g); it is likely due to the large intrinsic surface charge due to the high density of the hydroxyl groups of Fe_3_O_4_/APTES (air).

The aggregation and the disaggregation behavior of bare and silica-modified iron oxide nanoparticles samples at variable pH and variable concentrations of HA were investigated. Figure 7 shows the hydrodynamic diameter of iron oxide NPs as a function of pH in the presence of different HA contents. The increase of HA concentration increased MNPs aggregation, particularly at pH values close to the IEP, resulting in the formation of large, porous, fractal aggregates, which is in agreement with the DLVO theory and other data [56]. When 0.0077 g/g of humic acids was added, the average hydrodynamic diameter in the region of the isoelectric point for the samples varied from 329 for Fe_3_O_4_/APTES (Ar) to around 620 nm for Fe_3_O_4_/APTES (air). At high concentrations of HA, the Z average hydrodynamic diameter close to IEP shows clear trends: the larger size of aggregates with the higher concentration of HA (Figure 7, see Fe_3_O_4_/APTES (air)).

When the nanoparticle core is completely coated with humic acids, the average hydrodynamic size becomes 130–150 nm and does not depend on the value of pH. With IEP around pH~4 with the full-coverage amount of humic acids, the value of the hydrodynamic diameter for all samples is in the range of 150–200 nm (for Fe_3_O_4,_ this size observed in the pH range 4–10, for Fe_3_O_4_/APTES (air) in the range of 6–10).

Notably, HA superstructures limited the diffusion of both silica oligomers and nanoparticles in solution. Overall, these phenomena resulted in a smaller size of Fe_3_O_4_/APTES (Ar) than for the bare MNPs and Fe_3_O_4_/APTES (air) [57].

## 4. Conclusions

In this work, we have studied the Fe_3_O_4_/APTES multifunctional nanosorbents fabricated in different atmospheres (Ar and ambient) for environmental purposes, such as heavy metal or radioisotope removal by magnetic separation. The colloid stability of nanoadsorbents is essential since adsorption requires a well-dispersed state of NPs, while controlled aggregation is favored in the removal step. When focusing on the future applications, the microstructure and the colloid stability were investigated in terms of magnetite content, electrokinetic and hydrodynamic parameters and the pH-dependent loading capacity towards humic acids. It demonstrated that changes in the atmospheric conditions of the MNPs synthesis (in argon and ambient) led to a change both in the content and in the crystalline size of magnetite, as well as the surface charge, due to the variation in the ratio of surface functional groups (amino- and hydroxyl groups). Electrophoretic light scattering measurements performed at various HA loadings allowed us to observe the shift of the isoelectric point from pH~ 7 to 3 with increasing HA addition in parallel with the reversal of the zeta potential values. The latter occurred when the surface of the particle was completely covered by HA molecules, indicating that this phenomenon is closely related to HA adsorption. Based on the magnitude of the shift of the pH-dependent zeta potential functions due to the sorption of an increasing amount of HA observed for all samples, we can envision the construction of a series of samples in which the loading capacity varies. So, the zeta potential data can be used to predict the electrostatic sorption capacity (charge-dependent loading). Future studies will be directed on estimation the solubility of silica-coated magnetite nanosorbents functionalized by humic acids and their sorption capacity towards target ecotoxicants (heavy metals as Pb, Cu, pharmaceuticals as ciprofloxacin, diclofenac, streptomycine, paracetamol) and biomolecules.

## Figures and Tables

**Figure 1 nanomaterials-11-01418-f001:**
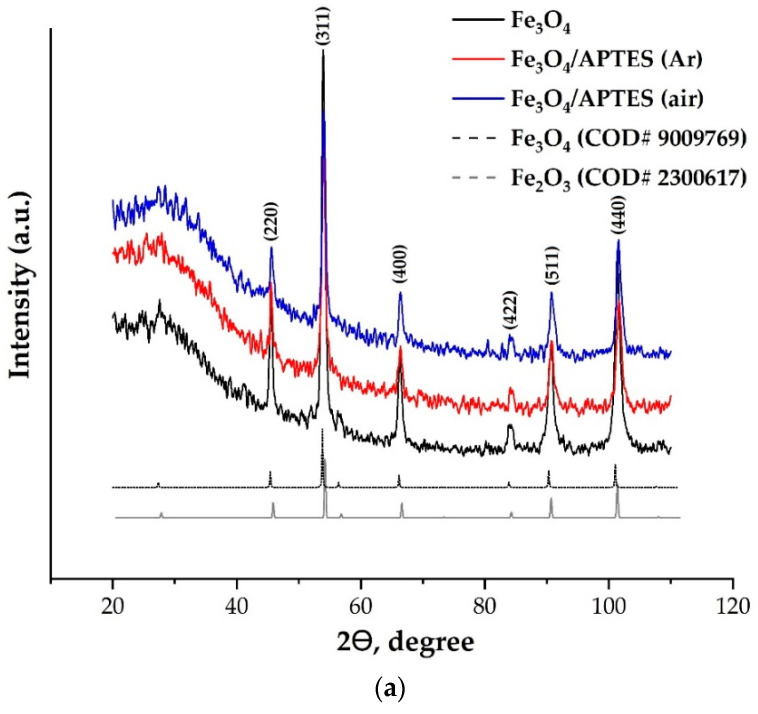
(**a**) Powder X-ray diffraction (XRD) of products with the magnetite and maghemite patterns from the Crystallography Open Database (COD) (**b**) Peak 440 of products and magnetite and maghemite COD patterns (**c**) Different models fitting by peak 440 of Fe_3_O_4_/APTES (air).

**Figure 2 nanomaterials-11-01418-f002:**
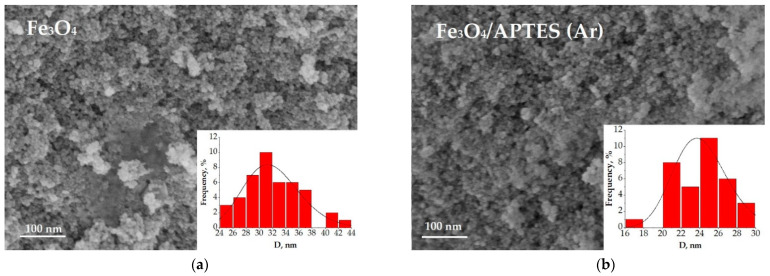
SEM image of (**a**) Fe_3_O_4_, (**b**) Fe_3_O_4_/APTES (Ar), (**c**) Fe_3_O_4_/APTES (air) and their size distribution.

**Figure 3 nanomaterials-11-01418-f003:**
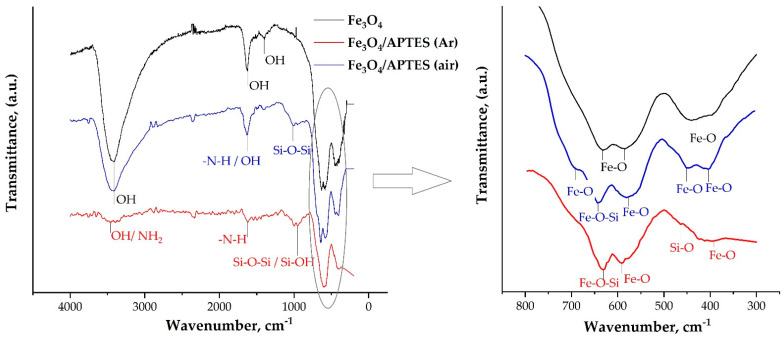
FTIR spectra of MNPs preparations.

**Figure 4 nanomaterials-11-01418-f004:**
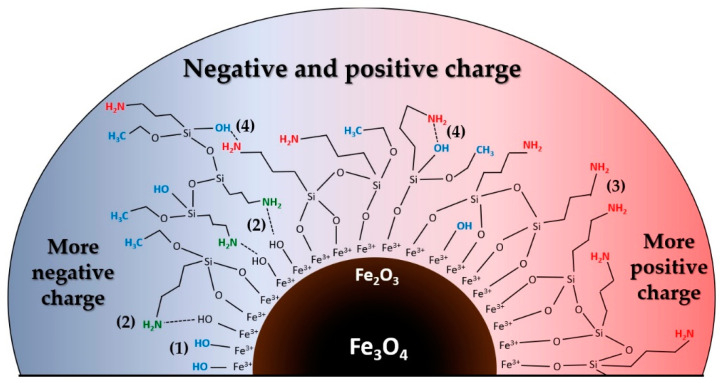
Principal scheme of Fe_3_O_4_/APTES with different types of bonds, which are possible on the surface: (**1**) Fe-OH in case of a lack of APTES due to unsuccessful functionalization (blue color on scheme); (**2**) C_2_H_5_OH (blue color) in case of an incomplete polymerization or ≡Fe-O-… + H_3_N-R bonds (green color on figure) or -NH_2_…-OH bands (**4**); (**3**) -NH_2_ in case of a successful functionalization (red color).

**Figure 5 nanomaterials-11-01418-f005:**
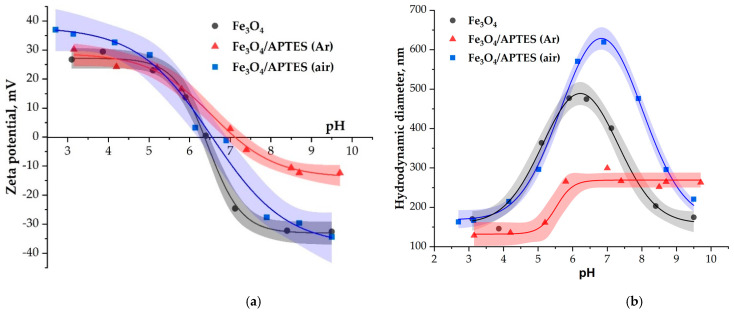
(**a**) Zeta potential and (**b**) hydrodynamic size of Fe_3_O_4_, Fe_3_O_4_/APTES (air) and Fe_3_O_4_/APTES(Ar) nanoparticles as a function of pH (0.01 M KCl). Value of zeta potential is given as ±0.2 of current pH.

**Figure 6 nanomaterials-11-01418-f006:**
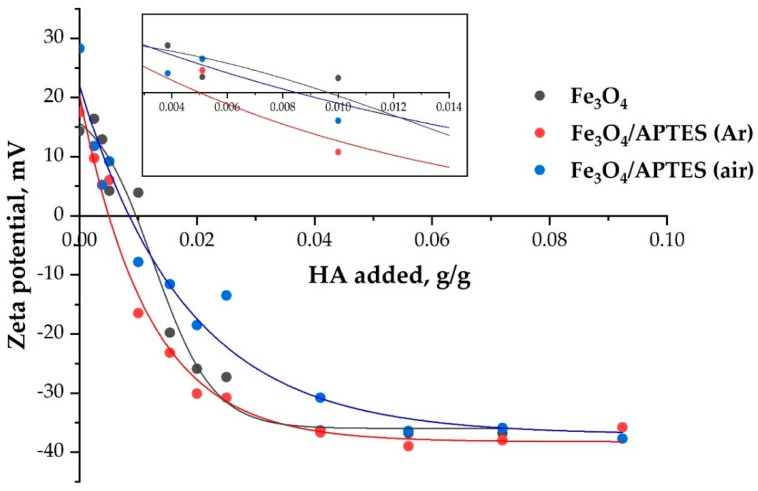
The effect of HA addition on the charge state of magnetite nanoparticles dispersions (pH ∼5 and KCl = 0.01 M). Lines are drawn as guides for the eye (*p*-value < 0.05).

**Figure 7 nanomaterials-11-01418-f007:**
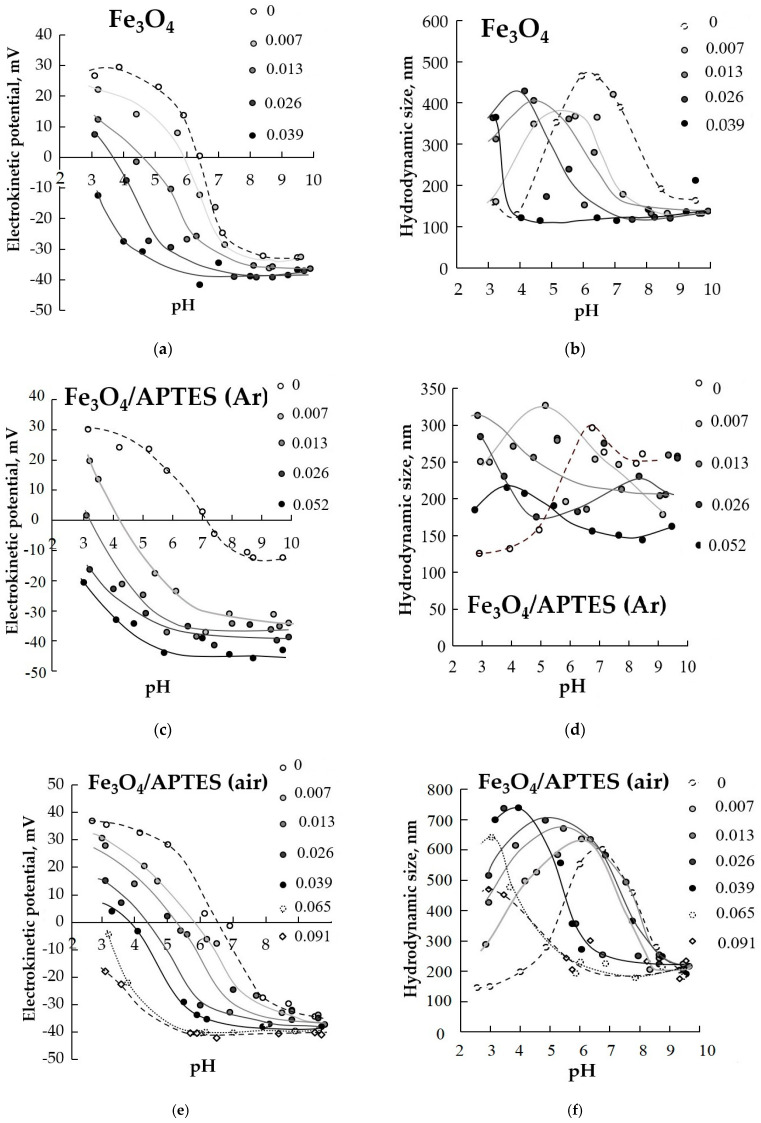
Effect of humic acids (g/g) adsorption on pH-dependent surface charging (**a**,**c**,**e**) and average hydrodynamic size of NPs (**b**,**d**,**f**) (zeta potential is proportional with the particle charge, 0.01 M KCl) (*p*-value < 0.05).

**Figure 8 nanomaterials-11-01418-f008:**
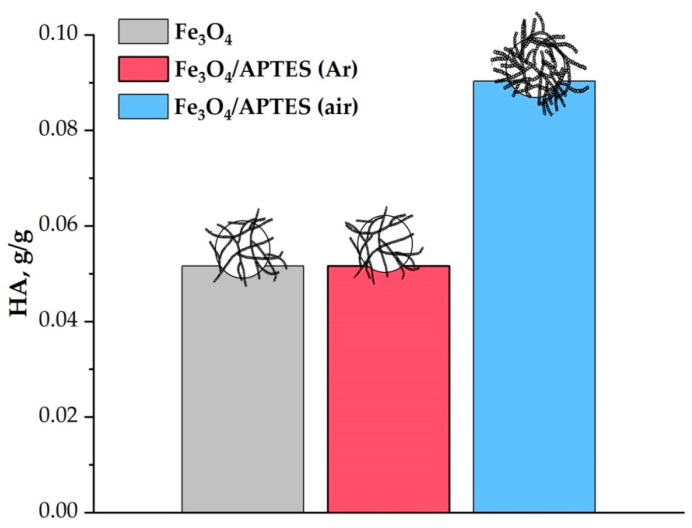
Amount of HA for complete coverage of 1 g NPs given in g. (pH 3, zeta potential~−20 mV and pH 9, zeta potential~−40 mV, 0.01 M KCl) (*p*-value < 0.05 for Fe_3_O_4_ or Fe_3_O_4_/APTES (Ar) and Fe_3_O_4_/APTES (air)).

**Figure 9 nanomaterials-11-01418-f009:**
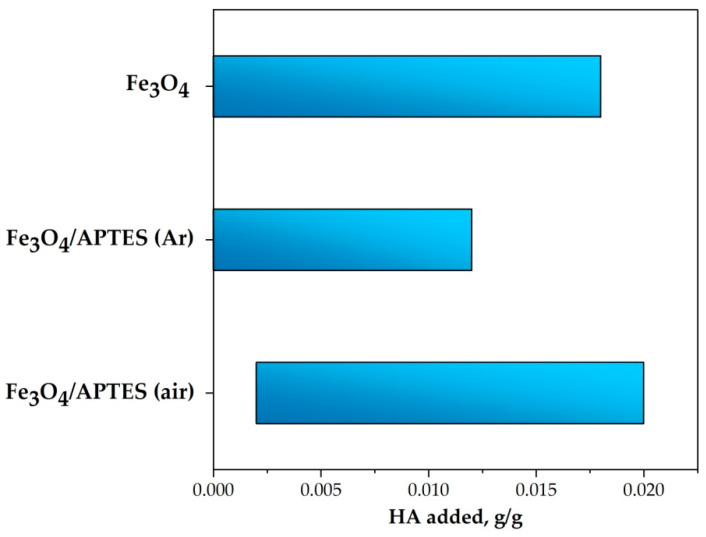
Range of amount of HA for unstable systems, g/g (pH ∼5 and KCl = 0.01 M) (*p*-value < 0.05).

**Table 1 nanomaterials-11-01418-t001:** Microstructure of MNPs.

Sample	Fe_3_O_4_			Fe_3_O_4_/APTES (Ar)			Fe_3_O_4_/APTES (Air)		
hkl	2θ°	d, Å	FWHM	2θ°	d, Å	FWHM	2θ°	d, Å	FWHM
220	45.45	2.965	0.636 (8)	45.45	2.96	0.504 (9)	45.65	2.956	0.721 (6)
311	53.90	2.527	0.662 (2)	53.95	2.53	0.679 (5)	54.05	2.519	1.731 (2)
400	66.30	2.095	0.780 (1)	66.3	2.095	0.890 (4)	66.4	2.089	0.822 (1)
422	83.85	1.714	0.975 (2)	84.05	1.712	0.844 (9)	84.45	1.707	0.785 (1)
511	90.70	1.610	0.940 (7)	90.75	1.61	0.828 (5)	90.7	1.609	1.891 (3)
440	101.35	1.481	0.899 (1)	101.37	1.481	1.026 (6)	101.55	1.477	0.824 (4)
a, Å		8.3813		8.3789			8.3603		
X		0.37		0.35			0.187		
δ		0.069		0.08			0.186		
Structure		Fe_2.93_O_4_		Fe_2.92_O_4_			Fe_2.81_O_4_		
% Fe_3_O_4_		78.8		75.8			42.4		
D_XRD_, nm		17.1 ± 2.3		20.5 ± 3.3			16.5 ± 1.96		
CV, %		13.5		16.1			9.5		
D_SEM_, nm		32.1 ± 4.3		24.2 ± 2.8			23.3 ± 3.1		
CV, %		13.5		11.6			10.5		

hkl—Miller indexes; *d*—interplanar distance, Å; *Q*—the angle at which the reflex was measured; FWHM—full width at half maximum of XRD reflex; *a*—interplanar distance, Å; *X*—the Fe^2+^/Fe^3+^ ratio; *δ—*calculated value, which ranges from zero (stoichiometric magnetite) to 1/3 (completely oxidized); D_XRD_—average particle size calculated by the Scherrer equation ± standard deviation, nm; CV—coefficient of variation characterizing the polydispersity of the system, %; D_SEM_—average particle size calculated by the SEM ± standard deviation.

**Table 2 nanomaterials-11-01418-t002:** Amount of HA needed to reach the changes in surface charge for the Fe_3_O_4_-APTES samples (pH ∼5 and KCl = 0.01 M).

HA Amount	Fe_3_O_4_	Fe_3_O_4_/APTES (Air)	Fe_3_O_4_/APTES (Ar)
for full neutralization of charge, g/g	0.01	0.0085	0.005
to reach −20 mV of zeta potential, g/g	0.016	0.038	0.014
to reach plateau, g/g	0.04	0.056	0.04

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
