# Peer review of "Fabrication, Microstructure and Colloidal Stability of Humic Acids Loaded Fe3O4/APTES Nanosorbents for Environmental Applications"

_nanomaterials, 2021, doi:10.3390/nano11061418_

Round 1

Reviewer 1 Report

The authors present in their draft: “Fabrication, microstructure and colloidal stability of humic acids loaded Fe3O4/APTES nanosorbents for environmental application” new approach for environmental purpose such as pollutants removal by magnetic separation.

It is a well done study.

But some of my comments should be considered before accepting for publication.

Introduction should be shortened about 30%.

The quality of SEM images should be substantially improved.

Figure 9 – why the pH value 5 was chosen for stability experiments?

Number of references is high, but well commented.

Real application of the developed nanosorbents for environmental purposes should be pointed out in conclusion section. This comment is missing.

Author Response

Dear Reviewer,

Thank you for your valuable and detailed comments regarding this article.

Please find below the revision’ details: notes and Reply/Correction.

Reviewer 2 Report

the work is interesting and well written. From my point of view it is a bit too technical and the authors, in such an extensive work, should focus more on the possibility of applying their results. For this I suggest adding some references such as, for example:

Calabrò, Francesco. "Modeling the effects of material chemistry on water flow enhancement in nanotube membranes." Mrs Bulletin 42.4 (2017): 289-293.

Barrera, Gabriele, et al. "Magnetic properties of nanocomposites." Applied Sciences 9.2 (2019): 212.

Mattia, Davide, Hannah Leese, and Francesco Calabrò. "Electro-osmotic flow enhancement in carbon nanotube membranes." Philosophical Transactions of the Royal Society A: Mathematical, Physical and Engineering Sciences 374.2060 (2016): 20150268.

Esposito, Serena, et al. "Separation of Biological Entities from Human Blood by Using Magnetic Nanocomposites Obtained from Zeolite Precursors." Molecules 25.8 (2020): 1803.

Author Response

(The authors gave the same response as above.)

Reviewer 3 Report

nanomaterials-1234002

Fabrication, microstructure and colloidal stability of humic acids loaded Fe3O4/APTES nanosorbents for environmental applications

Lyubov Bondarenko * , Erzsébet Illés , Etelka Tombácz , Gulzhian Dzhardimalieva , Nina Golubeva , Olga Tushavina , Yasuhisa Adachi , Kamila Kydralieva

Paper described series of silica-coated magnetite nanoparticles (MNPs) synthesized via sol-gel method with and without establishing an inert atmosphere and then tested in terms of humic acids (HA) loading applied as multifunctional coating agent. The influence of ambient conditions on the microstructure, colloidal stability and HA loading of different silica-coated MNPs has been established. The XRD , SEM, FTIR method were applied for MNPs characterization.

The strong aspects of the paper:

  • The topic of the paper is interesting from the cognitive point of view. A new silica-coated magnetite nanoparticles are synthetized.
  • The article contains elements of novelty and is well written.
  • The structure of the paper is correct.
  • The keywords are adequate and the abstract is informative.
  • The conclusion are based on the results.
  • The references are well selected.
  • The description of the obtained results is well done.

The weak aspects of the paper:

  • The figures need changes and correction. As was observed the Figures possess different size, different size font, some have a border others do not. The biggest problem is that they go beyond the margins. All Figures should be corrected to be similar. The figure size and font size should be unify.
  • References should be checked and the style will be unify e.g. when authors give number of page two style are present P. 546-552 or 546-552.
  • Line 322-331 – all chemical equations should possess own number.

I recommend minor revision of the paper.

Author Response

(The authors gave the same response as above.)
